# Propolis Extract and Its Bioactive Compounds—From Traditional to Modern Extraction Technologies

**DOI:** 10.3390/molecules26102930

**Published:** 2021-05-14

**Authors:** Jelena Šuran, Ivica Cepanec, Tomislav Mašek, Božo Radić, Saša Radić, Ivana Tlak Gajger, Josipa Vlainić

**Affiliations:** 1Department of Pharmacology and Toxicology, Faculty of Veterinary Medicine, University of Zagreb, Heinzelova 55, 10000 Zagreb, Croatia; jelena.suran@vef.hr; 2Director of Research & Development and CTO, Amelia Ltd., Zagorska 28, Bunjani, 10314 Kriz, Croatia; ivica.cepanec@amelia.hr; 3Department of Animal Nutrition and Dietetics, Faculty of Veterinary Medicine, University of Zagreb, Heinzelova 55, 10000 Zagreb, Croatia; tomislav.masek@vef.hr; 4Hedera Ltd., 4. Gardijske Brigade 35, 21311 Split, Croatia; bozo@hedera.hr (B.R.); sasa@hedera.hr (S.R.); 5Department for Biology and Pathology of Fish and Bees, Faculty of Veterinary Medicine, University of Zagreb, Heinzelova 55, 10000 Zagreb, Croatia; ivana.tlak@vef.hr; 6Division of Molecular Medicine, Ruđer Bošković Institute, Bijenička cesta 54, 10000 Zagreb, Croatia

**Keywords:** propolis, polyphenols, extraction

## Abstract

Propolis is a honeybee product known for its antioxidant, anti-inflammatory, anticancer, and antimicrobial effects. It is rich in bioactive molecules whose content varies depending on the botanical and geographical origin of propolis. These bioactive molecules have been studied individually and as a part of propolis extracts, as they can be used as representative markers for propolis standardization. Here, we compare the pharmacological effects of representative polyphenols and whole propolis extracts. Based on the literature data, polyphenols and extracts act by suppressing similar targets, from pro-inflammatory TNF/NF-κB to the pro-proliferative MAPK/ERK pathway. In addition, they activate similar antioxidant mechanisms of action, like Nrf2-ARE intracellular antioxidant pathway, and they all have antimicrobial activity. These similarities do not imply that we should attribute the action of propolis solely to the most representative compounds. Moreover, its pharmacological effects will depend on the efficacy of these compounds’ extraction. Thus, we also give an overview of different propolis extraction technologies, from traditional to modern ones, which are environmentally friendlier. These technologies belong to an open research area that needs further effective solutions in terms of well-standardized liquid and solid extracts, which would be reliable in their pharmacological effects, environmentally friendly, and sustainable for production.

## 1. Introduction

Propolis is one of the most known honeybee products, used in folk medicine since ancient times for its numerous health effects. Nowadays, it is a starting raw material for manufacturing various extracts that can be used as active pharmaceutical ingredients (APIs). Raw propolis is a natural, glue-like mixture collected by honeybees, mostly from flower and leaf buds of different plant species. In general, propolis consists of plant resins and essential oils, beeswax, and pollen. Organic compounds that have been identified in propolis are: polyphenols, terpenes, esters, amino acids, vitamins, minerals, and sugars [1,2]. Since plant material comes in contact with the honeybee’s digestive system, especially saliva, before being incorporated in the compartments of the beehive, propolis is considered to be an animal-derived product.

Honeybees use propolis primarily as a thermo isolation material by sealing the cracks in wooden walls and other parts of the hive and strengthening the construction of wax combs. Additionally, it is a crucial part of collective social immunity at the colony level.

The bioactive molecule profile of raw propolis varies according to the geographical and botanical origin, season, bees’ genetics, and environmental factors [3,4,5]. The quality and quantity of collected propolis depends on plant diversity and availability, source and term of gathering, beekeepers’ techniques and practices, as well as environmental health [6,7]. The special collector traps placed directly on frames and under the roof, or at the sides of a hive, are commonly used harvesting methods that ensure propolis extraction without contamination. Traps are nylon or plastic nets with small holes that stimulate worker bees to fill the propolis trap (Figure 1). Collected propolis is removed from frozen nets by flexing or brushing them. The raw material is usually ground up, sieved, and extracted using solvents such as ethanol (EtOH), glycol, or water [8].

The propolis extracts’ chemical profile will also depend on the extraction solvent type, solvent ratio, and extraction procedures. Overall, there are over 500 bioactive molecules identified in propolis, and most of them are secondary plant metabolites [9]. Many of these molecules have excellent antioxidant, anti-inflammatory, antimicrobial, immunomodulatory, antitumor, antiulcer, and wound healing effects [10,11,12,13,14]. They are studied individually and in various mixtures. Although polyphenols are a rather diverse group, they have many similarities and propolis pharmacology is mostly a result of their activity and interaction.

This review aims to present propolis both as the active pharmaceutical ingredient (API) and as a valuable source of other potential APIs. The latest research on the biological activity of propolis and its compounds outlines their commonalities and a multitude of effects. Since the activity of propolis will also depend on the extraction of its active compounds, we give an overview of extraction methods—from traditional to modern ones—with emphasis on improvements necessary for their application in pharmaceutical research and development.

## 2. Propolis Types, Key Molecules, and Their Biological Activities

The most abundant propolis type in Europe, Asia, and North America is poplar propolis, with plant source from *Populus* spp., mostly *P. nigra* L. Poplar propolis is rich with flavones, flavanones, phenolic acids, and their esters [3]. Birch propolis from Russia is also rich in flavones and flavonols but of a different type than poplar propolis [3]. In the tropical areas, the primary resource for green propolis from Brazil is *Baccharis* spp., especially *B. dracunculifolia*. This type is rich in diterpenoids, and prenylated phenylpropanoids, such as artepillin C (3,5-diprenyl-4-hidroxycinnamic acid; (**1**) and 3-prenylcinnamic acid allyl ester (**2**), which are used as green propolis markers (Figure 2). [15]. Brazilian red propolis is characterized by isoflavone formononetin (**3**), and isoliquiritigenin (**4**) (Figure 2) [4]. Cuban red propolis is rich in polyisoprenylated benzophenones like nemorosone (**5a**,**b**) (Figure 2) [16].

Bankova (1998) reported that Canarian propolis has a higher amount of furofuran lignans and sugars and sugar alcohols due to mucilaginous plants characteristic for the geographical area [17].

Geopropolis is a slightly different type of bee glue, a mixture of resins, wax, clay, or soil. It is produced by a stingless bee from genus *Melipona.* Dos Santos et al. (2017) determined polyphenols, flavanones, terpenoids, but also di- and trigalloyl and phenylpropanoid heteroside derivatives in hydroethanolic extracts of geopropolis [18], while the main phenolic compounds of geopropolis from Brazil were gallic (**6**) and ellagic acid (**7**) (Figure 3) [19].

The most abundant flavonoids found in propolis samples from Italy were chrysin (**8**), galangin (**9**), pinocembrin (**10**) and pinobanksin-3*O*-acetate (**11**). The most representative phenolic acids were caffeic acid (**12**), *p*-coumaric acid (**13**), and ferulic acid (**14**), as well as their derivatives, 3,4-dimethoxycaffeic acid (**15**; DMCA) and caffeic acid prenyl (**16**), benzyl (**17**), phenylethyl (**18**; CAPE), and cinnamyl (**19**) esters (Figure 4) [20]. This kind of polyphenol content is typical for poplar propolis.

Due to so many propolis types with different bioactive molecules, Bankova (2005) proposed the standardization according to the botanical origin and the corresponding chemical profile [3]. For poplar-type propolis, previously mentioned representative markers could be used for poplar-type propolis extract identification and standardization. The presence of markers in relevant concentrations should be used only for standardization and not as a direct measure of activity because it is impossible to attribute the activity of a complex mixture to a few components [3].

### 2.1. Molecular Mechanisms of Representative Propolis Markers Action

Polyphenols are secondary plant metabolites with an essential role in response to biotic stressors (plant pathogens, herbivores) and abiotic stress conditions, like drought and cold. Based on their molecular structure, they can be divided into flavonoids and nonflavonoids. Flavone chrysin, flavonol galangin, and flavanone pinocembrin are flavonoids, while most abundant nonflavonoids are phenolic acids subdivided into derivatives of benzoic acid, such as gallic acid (**6**) (Figure 3), protocatechuic acid (**20**) (Figure 5), and derivatives of cinnamic acid: caffeic (**12**), *p*-coumaric (**13**), and ferulic (**14**) acids (Figure 4) [21].

In general, polyphenols are best known for their antioxidant activity, and there is an ongoing debate about the mechanisms of their antioxidant action [22]. The simplest explanation is that, as free radical scavengers, they bind to radical oxygen species (ROS), like superoxide (•O_2_^−^), hydroxyl radical (•OH), and hydrogen peroxides (H_2_O_2_). Oxidative stress could be induced by various metals [23]. The binding affinity of polyphenols to metals leads to metal chelates formation, with increased antioxidant activity compared to parent polyphenol compounds [24]. Many of them, like kaempferol (**21**) (Figure 5), bind to zinc, a cofactor for more than 300 metalloenzymes essential for growth and development, and form very efficient radical scavenger complexes [25] having anticancer activity [26]. Polyphenols inhibit free radical generating enzymes like NADPH and xanthine oxidase [27] or increase the expression of antioxidant enzymes like superoxide dismutase (SOD) and catalase [28,29].

This happens via dissociation between Kelch-like ECH-associated protein 1 (Keap 1) and nuclear factor E2-related factor 2 (Nrf2) and activation of antioxidant response element (ARE) which is responsible for transcription of antioxidant/detoxification enzymes (Keap1/Nrf2/ARE pathway) [22]. Das et al. (2016) further elaborate on the antioxidant activity and relevance for many disease states via protein kinase C (PKC) regulation [22].

The target of polyphenol in its anticancer activity is a cell cycle by regulating phosphoinositide 3-kinase (PI3K)/Akt/mTOR signaling pathway [30]. They also inhibit pro-inflammatory factors, like the transcription factor nuclear factor kappa B (NF-κB) [31], cyclooxygenase-2 (COX-2), mitogen activated protein kinases (MAPKs), the production of TNF-α, interleukin-1-beta (IL-1-β), and IL-6 expression [32].

Their antimicrobial activity spans from the direct action against bacteria, viruses, and fungi, to suppression of microbial virulence factors, like biofilms. Many of them act synergistically with various antibiotics against multidrug-resistant microorganisms [21].

If we look closely into the poplar-type propolis representative polyphenols’ mechanisms of action [20] we will find many similarities.

### 2.2. Chrysin

In their review, Mani and Natesan (2018) describe numerous pharmacological effects of flavone chrysin (**8**) (Figure 4) [33]. The basis of chrysin’s organ-protective (e.g., neuroprotective, nephroprotective and cardioprotective) actions are antioxidant and anti-inflammatory effects, like suppression of redox-active transcription factor NF-kB [34], reduction of TNF-α [35] and IL-β generation [36], and inhibition of COX-2 and prostaglandin-E2 [37]. As Mani and Natesan (2018) mention, anticancer activity is linked to inhibition of angiogenesis [33], decreased cell proliferation, induction of cell death by apoptosis [38], and reduced inflammation [39]. As a Notch 1 activator, chrysin inhibits tumor growth [40]. It also inhibits human triple-negative breast cancer cells’ metastatic potential by modulating matrix metalloproteinase-10 (MMP-10), epithelial to mesenchymal transition, and PI3K/Akt signaling pathway [41].

Although chrysin’s antimicrobial activity is less in research focus, there is some potential as chrysin inhibits viral replication [42]. In contrast, its synthetic derivatives inhibit fatty acid biosynthesis (FAB) in *Escherichia coli*, *Pseudomonas aeruginosa*, and *Staphylococcus aureus* [43].

### 2.3. Galangin

Flavonol galangin (**9**) (Figure 4) also suppresses the inflammation by inhibiting NF-kB and PI3K/AKT signaling pathway [44,45]. Its antimetastatic activity is mediated through PKC/ERK signaling pathway [46] and ERK1/2 phosphorylation [47,48]. Antiproliferative activity of galangin and quercetin (also very abundant in propolis) was demonstrated in the human gastric cancer cell line (SGC-7901) where the apoptosis was induced via mitochondrial pathway involving caspase-8/Bid/Bax activation [49]. In their pioneer study, Pepeljnjak and Kosalec (2004) demonstrated the antibacterial activity of propolis ethanolic extracts (EEP) with consecutive isolation of bactericidal compounds with preparative chromatography, and by bioautography (bioassay in situ) they detected inhibition zones around galangin (**9**) [50]. The re-isolated galangin was further tested and expressed bactericidal activity against multiple-resistant bacteria: MRSA, *Enterococcus* spp., and clinical isolates of *P. aeruginosa.* Galangin (**9**) had an inhibitory effect on 16 strains of 4-quinolone resistant *S. aureus* [51]. The cytoplasmic membrane of bacteria is the target site for the activity of galangin as it disrupts its integrity producing loss of potassium, and the aggregation of bacterial cells [52,53]. Later, Ouyang et al. (2018) demonstrated that galangin (**9**) effectively inhibits murein hydrolase activity and the growth of vancomycin-intermediate *S. aureus* strain with the thickened cell wall, Mu50 [54]. Echeveria et al. (2017) claim that galangin (**9**) has better antimicrobial activity than quercetin due to well-spaced hydrophobic and hydrophilic regions in the molecule [55].

### 2.4. Pinocembrin

Flavanone pinocembrin (**10**) (Figure 4) has many similarities with the aforementioned compounds; for a review, see [56,57,58]. It increases the levels of superoxide dismutase (SOD) and glutathione, but decreases the levels malondialdehyde (MDA), myeloperoxidase (MPO) and ROS, nitric oxide, neuronal nitric oxide synthase (nNOS) as well as inducible NOS (iNOS) [59,60]. It downregulates PI3K/AKT, and NF-κB signaling pathways [61], ERK1/2 and Rho-associated protein kinase (ROCK) signaling pathways and Ca^2+^ concentration, and protects the mitochondria through ERK1/2-Nrf2 axis [62]. Reducing the content of Ca^2+^ in mitochondria prevents mitochondrial membrane swelling, ATP synthesis, and energy metabolism disorders caused by Ca^2+^ overload, and it also inhibits Mn-SOD activity [63,64,65,66]. Its neuroprotective activity is mediated through inhibition of p38 MAPK–MAPK-activated protein kinase-2–heat shock protein 27, and stress-activated protein kinase/c-Jun N-terminal kinase–c-Jun pathway [67], but it also conserves the ERK–cAMP-response element-binding protein (CREB)—brain-derived neurotrophic factor pathway (BDNF) [68]. Recently, pinocembrin was shown to be cardioprotective by enhancing glycolysis in the myocardium, which is an essential mechanism of action against ischemic injury of the heart [69]. It does so by promoting the expression of glycolytic enzyme 6-phosphofructo-2-kinase (PFKFB3) via transcription hypoxia-inducible factor (HIF)-1α [69]. Pinocembrin fatty acid acyl derivatives have antibacterial activity against *S. aureus* [70].

### 2.5. Nonflavonoids: Phenolic Acids

Nonflavonoid phenolic acids like caffeic (**12**), *p*-coumaric (**13**), and ferulic (**10**) have many effects in common with flavonoids and other propolis components (Figure 4).

Caffeic acid (**12**) is a noncompetitive inhibitor of PKC activity in partially purified human monocytes [71]. Although it exhibits anticancer effects, it can also protect cancer cells from oxidative stress and apoptosis [72]. Lin et al. (2012) showed that caffeic acid (**12**) attenuates apoptosis in non-small-cell lung cancer (NSCLC) cells via NF-κB, by upregulation of survival proteins survivin and Bcl-2 [72]. On the contrary, Min et al. (2018) reported synergistic effects of a high concentration of caffeic acid (**12**) (6× the concentration in the study of Lin et al., 2012) and paclitaxel in inducing apoptosis in NSCLC H1299 cells [73]. Caffeic acid-induced apoptosis of H1299 cells in a dose-dependent manner and a strong synergistic effect with paclitaxel was observed. It induced caspase-3 and -9 and increased expression levels of MAPK members, p-JNK, and p-ERK [73].

One of the most studied active compounds of a poplar-type propolis is caffeic acid phenethyl ester (CAPE) (18). It is a potent antioxidant extracted from propolis, with excellent anti-inflammatory, wound-healing, antidiabetic, organ protective, anticancer, and antimicrobial properties [74,75]. CAPE inhibits NF-κB and PI3/Akt and modulates MAPK pathways [74,75]. However, the effect on these pathways will depend on the cell type and probably on the concentrations of CAPE [76]. For example, in primary human CD4+ T cells, CAPE induced caspase-3 expression, inhibited NF-κB activation, protein kinase B (Aκt) phosphorylation, IFN-γ, and IL-5 secretion, with no effect on p38 MAPK phosphorylation [76]. On the other hand, in the neuropathic pain model, CAPE suppressed the phosphorylation of p38 MAPK, inhibited NF-κB, and decreased the expression of pro-inflammatory TNF-α, IL-1β, and IL-6 [77]. Thus, in both of these studies, CAPE had an anti-inflammatory effect.

CAPE’s antioxidant activity is mediated through Keap1/Nrf2/ARE pathway in the rat colitis model [78]. This mechanism of action is important in hyperglycemia, a state in which there is an increase in ROS and reactive nitrogen species (RNS), and the oxidative stress leads to the tissue damage [79]. CAPE is beneficial as an antioxidant and the inductor of heme oxygenase-1 (HO), Nrf2-regulated gene with a critical role in preventing vascular inflammation and mediated via p38 MAPK pathway [74,75,79]. The HO-1 induction results in cardioprotective effects in diabetes [80], neuroprotective in microglial cells [81] and nigral dopaminergic neurons [82]. In obesity and metabolic syndrome, the acute anti-inflammatory effect of CAPE on adipocytes [83,84] and upregulation of HO-1 could also be beneficial [85].

Wound-healing properties of CAPE result from antioxidant and anti-inflammatory effects but not in an early phase [86]. The study in mice bedsore model has shown that CAPE acts pro-inflammatory in the first three days by increasing mediators such as NOS2, TNF-α, and NF-κB, promoting macrophage migration and lipid peroxidation and decreasing Nrf2 expression [86]. However, all these effects were reversed seven days after the ulceration [86]. Thus, it seems that CAPE accelerates wound healing phases, from inflammatory to maturation, with long-term healing effects.

CAPE showed anticancer potential in numerous studies: from hematological, lymphoid, breast cancer, gastrointestinal, prostate, ovarian, cervical, head, and neck, to lung adenocarcinomas (for a review, see [75]). CAPE inhibits cancer cell growth by regulating the expression of tumor suppressor gene, N-myc downstream-regulated gene 1 (NDRG1) [87], even via several MAPK signaling pathways, and inhibition of STAT3 [88]. In addition, its inhibition of NF-κB in cancer cells induces apoptosis [89] and increases sensitivity to radiotherapy and chemotherapy [90]. The inhibition of PI3/Akt signaling pathway suppresses proliferation, induces cell cycle arrest and leads to apoptosis [91].

Neuroprotective effects of CAPE are mediated through already mentioned anti-inflammatory (mostly NF-κB inhibition) [75], antioxidant effects via MAPK and Akt/glycogen synthase kinase 3 (GSK3β) [92], MAPK and PI3/Akt pathway [93], NRF2/HO-1 [94], and JAK2/STAT3 pathway [95].

CAPE has antifungal, antibacterial, and antiviral activity. It prolongs the survival of mice [96] and *Caenorhabditis elegans* infected with *C. albicans* [97], and works synergistically with antifungal drugs like caspofungin, fluconazole [98], and amphotericin B [99]. CAPE is effective against Gram-positive bacteria like *S. aureus, Enterococcus faecalis*, and *Listeria monocytogenes* [100], bee infecting *Paenibacillus larvae* [101], common oral cariogenic bacteria (*Streptococcus mutans, Streptococcus sobrinus, Actinomyces viscosus*, and *Lactobacillus acidophilus*) [102,103], and Gram-negative *Vibrio cholerae* [104] and *E. coli* [105]. Besides the direct antibacterial effect, CAPE inhibits the bacterial virulence factors, such as the biofilm formation and development, the production of lactic acid and extracellular polysaccharides of *S. mutans* [102,103]. In addition, CAPE showed antiviral effects against the human immunodeficiency virus (HIV) [106], hepatitis C virus (HCV) [107], and type A and B influenza virus [108]. In more innovative forms like poly lactic-co-glycolic acid (PLGA) nanoparticles, CAPE was effective against *Leishmania* parasites [109].

Similarities with caffeic acid and CAPE can be seen in the biological activity of *p*-Coumaric acid (**13**) and ferulic acid (**14**) (Figure 4). *p*-Coumaric acid (**13**) decreases the production of iNOS, COX-2, IL-1β, a TNF-α [110] NF-κB, and pro-apoptotic proteins and increases Bcl-2 expression [111]. Ferulic acid (**14**) also inhibits the activity of NF-κB, expression of IL-6, and modulates the oxidative Nrf2 pathway [112].

### 2.6. Molecular Mechanisms of Propolis Extracts Action

Most of the effects mentioned above were studied for each compound separately. Still, it is crucial to consider the mixture effect of propolis since there are always synergisms and antagonisms between compounds in the mixture [113]. The mixture activity is not a mere addition of components activity but components interactions (synergisms and antagonisms) that can be challenging to predict. Although it was not possible to correlate the concentration of individual constituents with propolis biological activity [3] propolis extracts have shown activity comparable to those of its standard components.

For instance, caffeic acid and ethanolic extract of Brazilian propolis suppressed LPS-induced signaling pathways, like p38 MAPK, JNK1/2, and NF-κB in macrophages [114]. ERK1/2 was not affected by propolis extract [114]. An aqueous propolis extract inhibited macrophage apoptosis via glutathione (GSH) and the TNF/NF-κB pathway [115]. The ethanolic extract of Chinese poplar propolis protected vascular endothelial cells (VECs) from LPS-induced oxidative stress and inflammation, a result of inhibiting autophagy and MAPK/NF-κB signaling pathway, and reducing the phosphorylation of JNK, ERK1/2, and p38 MAPK [116].

It seems that no matter the origin and composition, propolis extracts will exert antioxidant action, but possibly through slightly different mechanisms of action. The EtOH extract of Brazilian red propolis (EERP) suppressed ROS generation and cytotoxicity by activating the Nrf2-ARE intracellular antioxidant pathway [117]. Zhang et al. (2016) reported that EtOH extracts of Chinese propolis (EECP) and EtOH extracts of *Eucalyptus* propolis (EEEP) improve antioxidant gene expression only via ERK/-Nrf2 signaling pathway, while EtOH extracts of *Baccharis* propolis (EEBGP) strengthen the antioxidant system by activating p38 MAPK and accelerating nucleus translocation of Nrf2 [118].

Propolis extracts are active against yeasts, fungi, viruses, bacteria, and even parasites, being the most effective against Gram-positive bacteria, such as *Streptococcus* and *S. aureus*, *Bacillus subtilis*, *E. faecalis*, and against yeasts of *Candida* species [119,120]. The antimicrobial activity is often prescribed to its polyphenols that increase bacterial membrane permeability, disturb the membrane potential, reduce ATP production, and decrease bacterial mobility [2]. Some of them damage the biofilm and have anti-quorum sensing activity [121].

Most probably, the activity of polyphenols in propolis should be attributed to interactions between them and other components and follows the Goldilocks principle. The other, not yet identified compounds like peptides could also affect polyphenols’ activity and stability [113,122]. However, the most important is the issue of polyphenol bioavailability, which seems to be very variable—the most abundant molecules may not have an adequate profile [123]. The relevant factor to affect the bioavailability would undoubtedly be the method of propolis extraction and standardization [124]. As the knowledge of propolis (mixture and components) pharmacology grows, so do extract preparation technologies.

Interestingly, although propolis content varies depending on extraction, it seems like different extract’s biological activities remain comparable. Galeotti et al. (2018) demonstrated that propolis solubilized in various solvents and liquid and solid forms has a similar chemical composition when produced from the same raw material, with differences in total polyphenol content but comparable antioxidant activity [8]. Mašek et al. (2018) reported the dependence of extract chemical profile on solvent ratio and extraction procedures [125]. Maceration gave the highest quantity of aromatic acids, while microwave-assisted extraction led to the extraction of the highest flavonoids. However, these differences did not significantly affect antimicrobial activity [125].

## 3. Technologies for Propolis Extraction and Types of Propolis Extracts

The crude propolis is traditionally extracted with extraction solvents (ES) consisting of various EtOH and water mixtures. Typically, 25–60% *v*/*v* aqueous EtOH is used as the ES at room temperature (r.t.), yielding the propolis tincture. Such a simple extraction process is known as maceration. It is conducted by the addition of aqueous EtOH onto the crude propolis chunks at the weight (propolis) to volume (ES) ratio of 1:3–20, most commonly 1:5–10. This obtained mixture is usually left to stand in a closed vessel at r.t. for 7–30 days. In a typical example of such procedure, the crude propolis is subjected to a r.t. maceration process for 30 days [126]. Usually, the most optimal maceration time is about 10 days. A prolonged maceration time of 20 or 30 days will result in a very slight increase of polyphenols yield in the resulting liquid extract [126]. After the maceration, the liquid extract is separated from undissolved propolis residues by simple filtration. Thus, obtained dark brown liquid is commonly used as an active pharmaceutical ingredient (API), active cosmetic ingredient (ACI), or functional food ingredient (FFI) in the production of various pharmaceutical, or cosmetic products. In its simplest version, its strength is defined by the so-called drug-to-extract (DER) ratio, which represents a weight ratio of starting crude propolis against the final liquid extract obtained by such a method. For instance, if 100 g of liquid extract is manufactured from 10 g of starting propolis, then the DER ratio of such extract is 1:10.

Park and coworkers (1998) described the extraction of the crude propolis with various mixtures of water (5–100%) and EtOH (0–95%) at 70 °C for 30 min [127]. After cooling and filtration of undissolved residue, obtained liquid propolis extracts were analyzed for their antimicrobial activity against *S. aureus* as a model pathogenic microorganism, antioxidant effect, and inhibition of hyaluronidase activity. The results showed that the most active extracts were those obtained with 60–80% aqueous EtOH, while those with 40–60% and 80–95% EtOH showed significantly lower activities in all three comparative pharmacological activities [127]. Obviously, lower EtOH content in the ES resulted in poorer extraction ability of the corresponding EtOH-H2O mixture and accompanied lower content of active propolis ingredients, which resulted in comparably decreased pharmacological effects.

Several disadvantages characterize such liquid propolis extracts:(i)The presence of a relatively aggressive solvent (EtOH);(ii)Alcohol-based products are not suitable for children, pregnant and breastfeeding women, and certain patients;(iii)And relatively high content of beeswax, which causes its separation upon the phase of mixing with water phase, during the manufacturing of pharmaceutical and other products, where such an extract is employed as an API.

Due to these deficiencies of such traditional liquid propolis extracts, various improved solutions are developed and described in numerous scientific and patent literatures. Some selected, typical improved solutions are presented in this article.

Except by a simple maceration process in a single extraction vessel, optionally equipped with a suitable stirrer for adequate agitation of the propolis mass during the maceration process, the extraction can be performed by percolation or with a Soxhlet apparatus [128]. These techniques significantly speed up the extraction process and somewhat increase the extraction efficiency expressed through the quantitative content of certain marker propolis components.

Certain progress has been introduced by the use of pure water as a sole extraction system [128,129,130,131,132,133]. Water is more polar solvent than 60–80% aqueous EtOH, and it extracts more polar propolis compounds. The efficacy of water as the propolis ES could be increased by the use of:(i)Repeated extraction at cold processing conditions [131];(ii)Elevated extraction temperatures ranging from 50–95 °C [129,130,132,133];(iii)Soxhlet extraction technique [128];(iv)Or ultrasonic-assisted extraction (UAE) [133].

Sosonowski described the preparation of the liquid propolis extracts obtained by maceration of the crude propolis with various alternative organic solvents (OS) such as: methanol (MeOH), 1-propanol (*n*-PrOH), 2-propanol (2-PrOH), 1-butanol (*n*-BuOH), 2-butanol (*s*-BuOH), tert-butanol (*t*-BuOH), diethylether (Et_2_O), 1,2-propylene glycol (1,2-PG), dimethylsulfoxide (DMSO), ethylene glycol (ETG), benzyl benzoate (BnBz), polyethylene glycol (PEG), acetone, and glacial acetic acid (HOAc) at a low (1:2) ES-to-propolis weight ratio (*w*/*w*) [134]. Such liquid extracts were optionally converted to dry extracts by evaporation of the respective solvents at elevated temperatures (70 °C) under vacuum.

Among more polar extraction solvents, especially convenient are 1,2-PG, PEG (200, 400, or 600), or glycerol (GL), which are relatively non-toxic, safe, and widely used as pharmaceutical excipients or as diluents and humectants in numerous pharmaceutical products [8,135,136].

1,2-PG was found to be very effective propolis ES, giving high percentage of extracted polyphenols at DER ratio 1:10 to 1:20 at 50–60 °C for a short period of time (2 h), or during the maceration at r.t. for several days [135]. PEG 400 is a comparably good ES for propolis extraction. Its mixtures (e.g., 20% *w*/*w*) with water resulted in significantly increased total phenolic content (TPC) in comparison to pure aqueous propolis extracts [136]. When compared to 1,2-PG, GL is more polar and viscous solvent that causes certain difficulties during the extraction process. Its higher polarity enables preferential extraction of relatively more polar propolis ingredients. The relatively high viscosity diminishes its penetration ability to the propolis mass, which results in decreased extraction efficacy at r.t. This could be overcome by processing the extraction mixture at elevated temperatures. Thus, Galeotti and coworkers (2018) described that GL is useful as the ES for propolis with the comparable parameter of an antioxidant capacity (μg Trolox equivalents per mg of polyphenols) of resulting liquid extract [8]. Still, the polyphenols content was significantly lower than at traditional hydroalcoholic and 1,2-PG extractions [8].

Additionally, Tsukada and co-workers (1991) disclosed the use of GL as the ES at very low weight ratio of propolis-to-ES, 1:2 *w*/*w*, at 90–160 °C, with subsequent filtration. Such glycerol extracts are water-soluble and suitable as API for the production of various pharmaceutical products [129].

The least polar ES that could be employed for the propolis extraction are various fixed and essential oils. The former are various triglycerides, plant oils such as sunflower, rapeseed, sesame, olive, and soybean oils. They are able to extract predominantly non-polar propolis compounds from the starting crude propolis. In this manner, Galeotti and coworkers (2018) described the extraction with plant oil and obtained the corresponding liquid extract of similar TPC like with GL as the ES [8]. This type of extract is suitable for manufacturing various pharmaceutical and cosmetic products containing fatty phase, wherein such lipophilic ingredients could be readily homogenized.

Similarly, Savickas and coworkers (2001) described a unique waterless ES based on a mixture of EtOH (96%) and sunflower oil [137]. Such a liquid propolis extract is also suitable for non-aqueous pharmaceutical and cosmetic formulations.

Keskin (2020) described the use of orange peel essential oil (OPEO) predominantly based on d-limonene as a green and renewable ES for propolis [138]. The extraction was performed at 1:10 *w*/*v* ratio of the crude propolis to the ES during 48 h at r.t. with constant stirring. After subsequent filtration, the resulting liquid extract was of a high total phenolic content (mg of gallic acid equivalents per mL) and high total flavonoid content (mg). The former parameter was about 50% lower, while the latter was around 28% lower than at 70% EtOH-based extract [138].

Further progress has been made in the field of the utilization of natural deep eutectic solvents (NADES), which exhibit good solvent properties, while safe, of low toxicity, and from renewable resources [139,140]. Thus, de Funari and coworkers (2019) examined several NADES systems in propolis extractions and found that:(i)The choline chloride (CC):1,2-PG in a molar ratio (n:n) 1:1 or 1:2;(ii)CC:lactic acid (LA):H_2_O in 1:2:2 or 1:1:1, n:n:n;are very effective ESs for propolis extraction, comparable to aqueous ethanol (70%) as a golden standard ES [140]. Alternatively, aqueous solution of amino acid *L*-lysine (Lys; 10%) is relatively effective ES for propolis, yielding the corresponding liquid extract with about 50% of the component’s concentration obtained with aqueous EtOH (70%) [139,140].

Modern green-chemistry technology for the manufacturing of solid propolis extracts includes supercritical carbon dioxide (scCO_2_) as the extraction solvent [141]. When exposed to a high pressure (>7.4 MPa) and temperatures higher than 31.1 °C, CO_2_ is transformed into a supercritical fluid, whose solvent properties are roughly similar to acetone. It is able to dissolve different non-polar to slightly polar propolis compounds. After this supercritical extraction (scE), the liquid scCO_2_ propolis extract is subjected to the filtration within the closed system to remove undissolved propolis residues [141]. The resulting liquid scCO_2_ propolis extract is simply converted to a dry powdery propolis extract by evaporation (and regeneration) of CO_2_ by cooling the system and reducing the crystallization unit’s pressure [141]. The main advantage of scCO_2_ extraction is relatively mild extraction conditions, easy removal of the ES, no residual solvent impurity, and a highly positive environmental impact due to the complete absence of organic solvents [141].

Another unique approach to propolis extraction is applying reagents dissolved in the ES that are capable of complexing low molecular weight (M_w_) propolis components by forming host-guest molecular complexes. A typical example of such a complexing agent is hydroxypropyl-β-cyclodextrin (HP-β-CD). The latter facilitates the propolis extraction via decreasing the equilibrium concentration of low M_w_ propolis compounds within the supernatant liquid phase of the extraction suspension by their complexation into the hydrophobic HP-β-CD cavity. This causes the dissolution of a higher amount of propolis compounds of relatively lower polarity and thus increases the extraction process’s overall efficacy [142]. Typical solvent system is based on an aqueous GL (30–70% GL) with 10–30% HP-β-CD [142]. Thus, obtained liquid propolis extract is suitable for manufacturing various pharmaceutical, cosmetic, and food products, where no further removal of the ES is required. Since all ES components are edible or compatible with many pharmaceutical formulations, their presence does not represent any difficulty for practical utilization of such liquid propolis extracts.

Pellati and coworkers (2013) described ultrasonic-assisted (UAE) and microwave (MW)-assisted (MAE) extractions of propolis with a mixture of EtOH and water (80:20, *v*/*v*), and compared their efficiencies with a traditional maceration process and conventional heat reflux extraction (HRE) [20]. Their results demonstrate that both these modern technologies result in a comparable extraction efficacy compared to traditional extraction technologies under similar or shorter processing time [20].

Trusheva and coworkers (2007) described that the use of UAE and specially MAE do result in dramatically reduced extraction processing time with proportional extraction efficiency and somewhat higher chemical yields [143]. For instance, the propolis extraction with traditional aqueous EtOH (75% *v*/*v*) under MAE conditions (800 W; 2 × 10 s) resulted in approximate extraction efficacy and higher (73–75%) yield when compared with UAE that took 30 min yielding 41–53%, or with traditional maceration that required 72 h and gave 55–58% yield [143].

In contrast to these results, Oroian and coworkers (2020) described that UAE was more efficient in terms of the chemical yield of the extraction and higher content of active propolis substances against the corresponding products obtained with traditional maceration and MAE. Perhaps 2 × 1 min of their MW (2450 MHz) at 140 W was too mild for more effective extraction than 2 × 15 min duration of UAE (20 kHz) process [144].

An additional modern approach to the propolis extraction is based on a high-pressure extraction (HPE; or high hydrostatic pressure extraction, HHPE), in a batch-type high-pressure extraction vessel, which stimulates the transport of ES into the solid crude propolis particles [145,146]. This results in dramatically reduced extraction time and comparable to higher yields of extracted active propolis ingredients. Jun described the propolis extraction at 500 MPa with 75% aqueous EtOH, yielding the corresponding liquid extract with comparable antioxidant activity as the control extract obtained by a room temperature maceration process [147]. However, compared to the traditional maceration that required 7 days, the HPE was accomplished within just 1 min [147].

The use of suitable emulsifiers introduced further significant improvements in environmentally friendly, water-based technologies. They facilitate the solubilization of relatively less polar propolis active substances into the water ES medium. In this manner, Paradkar and coworkers (2010) described the process for the propolis extraction by the use of an aqueous polysorbate (PS) solution at 40–90 °C during 2–24 h, yielding the corresponding liquid propolis extract [148]. PSs are widely used food- and pharma-grade emulsifiers based on higher fatty acid esters of ethoxylated sorbitan. Their hydrophilic–lipophilic balance (H.L.B.) is roughly between 14–18. It can facilitate the extraction process via the formation of micelles, which solubilize relatively non-polar propolis active substances into their intramicellar cavities. Beside PSs, various other types of emulsifiers, polymers, and plant gums were used as the solubilization enhancing agent: polyoxyethylene castor oil derivatives (PECO); lecithin (LE); monoglycerides (MG) such as glyceryl monooleate, monostearate or palmitostearate; polyvinyl alcohol (PVA); polyethylene oxide (PEO); simple and cross-linked polyacrylamide (PAA/cPAA); sodium carboxymethylcellulose (CMC); guar gum (GG); xanthan gum (XG); and cross-linked polyacrylate (cPA) and some other solubilizers [148].

In this field, Radic et al. (2019) described a unique non-aqueous ES system based on the solution of plant lecithin (LE), e.g., rapeseed lecithin, in liquid polyethylene glycol such as PEG 400, which effectively extracts the crude propolis at r.t. or at elevated temperatures (up to 150 °C) yielding the corresponding liquid extract (after appropriate filtration), which is suitable as the ACI or API for the production of various cosmetic, pharmaceutical, and veterinary medicinal products [149]. This product has been standardized to the content of four main targeted propolis active ingredients: 100–1300 μg/mL *p*-coumaric acid (**9**), 75–800 μg/mL *trans*-ferulic acid (**10**), 25–300 μg/mL caffeic acid (**8**), and 40–400 μg/mL 2-phenylethyl 3,4-dihydroxycinnamate (**14**) [149].

There is a vacuum resistive heating extraction (VRHE) process among special new technologies, which is based on a vacuum Ohmic heating process [150]. Lastriyanto and Kartika (2020) described the two-step propolis extraction with water as the ES in the first and 70% aqueous EtOH in the second step by exposing the extraction mixture to an electric current (100–220 V) under vacuum (16.6 kPa) at 37–58 °C. The most profound improvement with the VRHE technology is significantly enhanced extraction efficacy, e.g., the gallic acid equivalent (GAE) of the resulting liquid propolis extract was 45.72 mg GAE/g. In comparison, the control product obtained by the traditional maceration product (after 50 °C/24 h extraction) had only 24.21 mg GAE/g [150]. The total flavonoid content was approximately 5× higher than in the product obtained with the maceration process [150].

Described selected typical technologies for propolis extraction are given in Table 1.

### Preparation of Solid Propolis Extracts

The preparation of powdery propolis extracts usually starts from suitable primary liquid extracts, obtained with volatile ES such as water, aqueous EtOH (40–80% *v*/*v*), EtOH (96% *v*/*v*), 2-PrOH, acetone, or similar low toxic organic solvents. The latter are usually further processed through the spray-drying or fluid bed technology onto a suitable powdery carrier such as food- or pharma-grade maize-, wheat- or potato-based maltodextrin, with or without subsequent addition of a small (0.2–2% *w*/*w*) amount of magnesium stearate, colloidal silicon dioxide, or similar anti-caking agent. Alternatively, powdery propolis extracts can be prepared directly via scCO_2_-based extraction [141,146]. The spray-drying technology could be adjusted in a manner to produce a solid extract that is coated with suitable food- or pharma-grade inert powdery materials, carrier, and coating excipients. In this case the process is called encapsulation [151,152,153]. This technology provides significantly higher chemical stability and increased physical stability against humidification and resulting caking, as is the case with simple solid extracts adsorbed on maltodextrin [151]. An optional technology is a freeze-drying, which enables additional mildness to the extraction and thus preserves more thermal sensitive active ingredients [152]. Beside maltodextrin, other suitable food- and pharma-grade coating agents including gum Arabic [151,152], inulin [152], chitosan-DNA [153], and chitosan-pectin [153] could be employed.

## 4. Conclusions

There are various types of propolis, with different compositions and activities. Pharmacological effects of propolis extracts are very similar to the effects of their bioactive molecules’, although they result from the interaction between them. Thus, the composition and activity of extracts will depend on propolis’s botanical and geographical origin and the method of extraction. More polar solvents will lead to better extraction of polar molecules, while non-polar organic and oily solvents will extract non-polar molecules. Organic polar solvents will enable a good extraction yield of both non-polar and polar molecules. Nevertheless, all propolis extracts will have antioxidant, antimicrobial, and anti-inflammatory activity. Still, extract standardization will be crucial in obtaining reproducible pharmacology, essential in developing propolis extract as API.

To better describe propolis extraction technologies’ characteristics, we overviewed more traditional ones to modern and environmentally friendly processes. In contrast to existing reviews, the special attention was paid to environmentally friendly extraction processes, non-toxic ES (H_2_O, PEG), and additives that enhances extraction chemoselectivity such as emulsifiers (LE). The propolis extraction technologies still belong to an open research area that needs further effective solutions in terms of well-standardized liquid and solid extracts, which would be reliable in their pharmacological effects and environmentally friendly and sustainable strategies for their production.

## Figures and Tables

**Figure 1 molecules-26-02930-f001:**
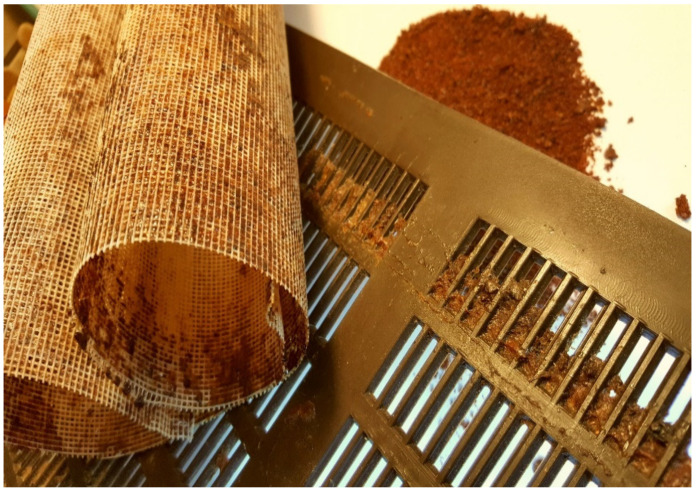
Two types of collector traps and harvested raw propolis.

**Figure 2 molecules-26-02930-f002:**
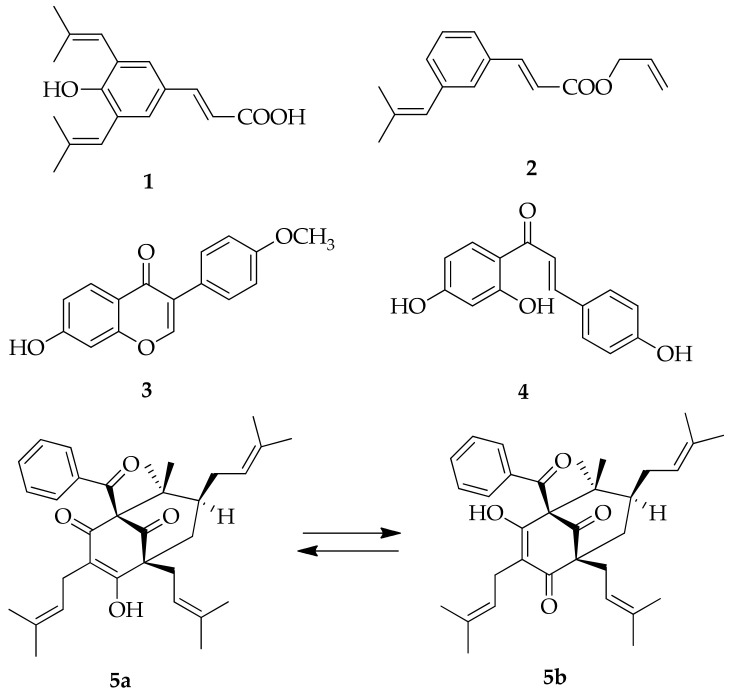
Molecular structures of representative Brazilian and Cuban propolis markers: 3,5-diprenyl-4-hidroxycinnamic acid (**1**), 3-prenylcinnamic acid allyl ester (**2**), formononetin (**3**), isoliquiritigenin (**4**), nemorosone (**5a**,**b**).

**Figure 3 molecules-26-02930-f003:**
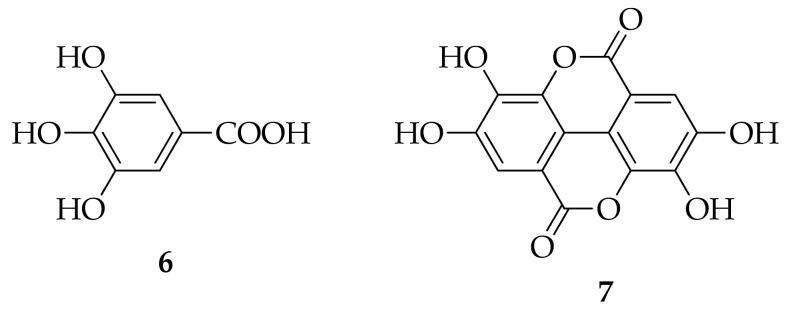
Molecular structures of representative Brazilian geopropolis markers: gallic acid (**6**), ellagic acid (**7**).

**Figure 4 molecules-26-02930-f004:**
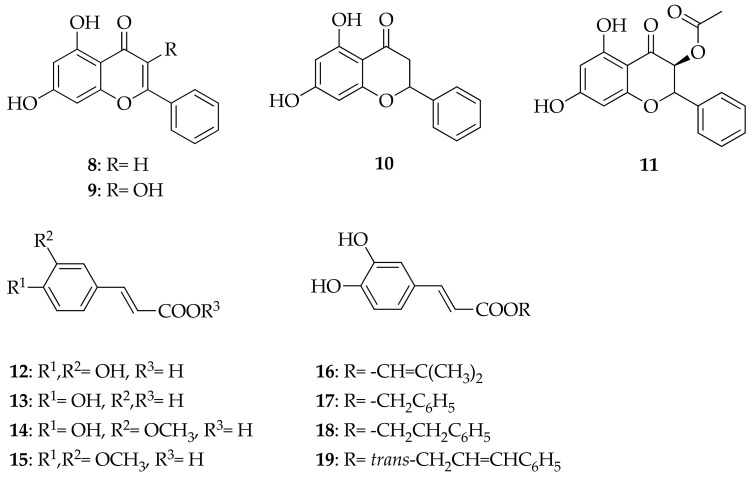
Molecular structures of representative poplar propolis markers: chrysin (**8**), galangin (**9**), pinocembrin (**10**), pinobanksin-3*O*-acetate (**11**), caffeic acid (**12**), *p*-coumaric acid (**13**), ferulic acid (**14**), 3,4-dimethoxycaffeic acid (**15**; DMCA), caffeic acid prenyl (**16**), benzyl (**17**), phenylethyl (**18**; CAPE), cinnamyl esters (**19**).

**Figure 5 molecules-26-02930-f005:**
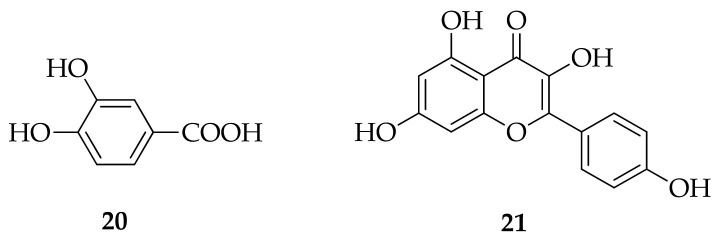
Molecular structures of protocatechuic acid (**20**) and kaempferol (**21**).

**Table 1 molecules-26-02930-t001:** Extraction technologies for crude propolis extraction.

No	Extraction Solvent (ES) (Special Conditions)	Extraction Type	P:ES Ratio (*w*/*v*)	Extraction Temperature	Extraction Duration	Ref.
1	EtOH (25–60% *v*/*v*)/H_2_O	M	1:5	r.t.	typically 10–30 days	[126]
2	EtOH (10–95% *v*/*v*)/H_2_O	M + P, SE	1:12.5	Δ/70 °C	30 min	[127,128]
3	H_2_O	E	n.r.	Δ/30–50 °C	6–8 min	[129]
4	H_2_O	E	1:20	Δ/70–95 °C	18 h	[130]
5	H_2_O (4× repeated with fresh H_2_O)	M	1:4	r.t.	72 h	[131]
6	H_2_O	E	1:2	Δ/60 °C	30 min	[132]
7	H_2_O	UAE	1:10	Δ/50–60 °C	2.5 h	[133]
8	OS: MeOH, *n*-PrOH, *i*-PrOH, *n*-BuOH, *s*-BuOH, *t*-BuOH, Et_2_O, BnOH, 1,2-PG, DMSO, ETG, BnBz, PEG, acetone, HOAc	M	1:2	r.t.	10 days	[134]
9	1,2-PG	M/E	1:10–1:20	r.t. or Δ/50–60 °C	10 days 2 h	[135]
10	H_2_O/PEG 400 (20% *v*/*v*)	M	1:10	r.t.	5 h	[136]
11	Glycerol (GL)	n.r.	n.r.	n.r.	n.r.	[8]
12	Glycerol (GL)	E	1:2	Δ/90–160 °C	n.r.	[129]
13	Plant oil	n.r.	n.r.	n.r.	n.r.	[8]
14	EtOH (96%)/sunflower oil (60:40 *w*/*w*)	n.r.	n.r	n.r	n.r	[137]
15	OPEO (mostly d-limonene)	M	1:9	r.t.	48 h	[138]
16	CC/1,2-PG (1:1, n/n)	E	1:20	Δ/50 °C	3 h	[139,140]
17	CC/1,2-PG (1:2, n/n)	E	1:20	Δ/50 °C	3 h	[139,140]
18	CC/LA/H_2_O (1:2:2, n/n/n)	E	1:20	Δ/50 °C	3 h	[139,140]
19	CC/LA/H_2_O (1:1:1, n/n/n)	E	1:20	Δ/50 °C	3 h	[139,140]
20	Lys/H_2_O (1:10, n/n)	E	1:20	Δ/50 °C	3 h	[139,140]
21	scCO_2_	scE	1:10	Δ/40–60 °C 10–20 MPa	n.r.	[141]
22	H_2_O/GL (3:1)/HP-β-CD (22.5%) or H_2_O/GL (1:1)/HP-β-CD (11.25%)	CAAE	n.r.	n.r.	n.r.	[142]
23	EtOH/H_2_O, 80:20, *v*/*v* (UAE; 120 W; closed vessel)	UAE	1:10	Δ/70 °C	1 h	[20]
24	EtOH/H_2_O, 80:20, *v*/*v* MAE: 300 W/2450 MHz/ closed vessel	MAE	1:10	Δ/106 °C	15 min	[20]
26	EtOH/H_2_O, 70:30, *v*/*v* UAE: 300 W/20 KHz MAE: 800 W/2450 MHz	UAE MAE	1:10 or 1:20	25 °C	UAE: 30 min MAE: 2 × 10 s	[143]
27	EtOH/H_2_O, 70:30, *v*/*v* UAE: 20 KHz MAE: 140 W/2450 MHz	UAE MAE	1:50	UAE: r.t. MAE: ≈60 °C	UAE: 15 min MAE: 2 × 1 min	[144]
28	EtOH/H_2_O, 75:25, *v*/*v* HPE: 500 MPa	HPE	1:35	r.t.	1 min	[147]
29	H_2_O/EM EM: PS, PECO, LE, MG, PVA, PEO, PAA, cPAA, CMC, GG, XG, cPA	E	typically 1:10	Δ/40–100 °C	2–24 h	[148]
30	PEG200–600/LE (0.1–3.5% *w*/*w*)	M/E	1:2–1:20	r.t. or 10–150 °C	5 min to 72 h	[149]
31	1. step: H_2_O/16.6 kPa/100 V 2. step: EtOH/H_2_O, 70:30, *v*/*v*, 16.6 kPa/220 V	VRHE	n.r.	1. 58 °C 2. 37 °C	2 × 20 min	[150]

P:ES ratio = a ratio between the weight starting crude propolis and volume of employed extraction solvent, expressed as *w*/*v*; M = maceration, a batch-type extraction at room temperature (r.t.; typically 20–25 °C); P= percolation; SE = Soxhlet extraction; E = a batch-type extraction at elevated temperatures; UAE = ultrasound-assisted extraction; OS = organic solvent; MeOH = methanol; *n*-PrOH = 1-propanol; *i*-PrOH = 2-propanol; *n*-BuOH = 1-butanol; *s*-BuOH = 2-butanol; *t*-BuOH = *tert*-butanol; Et_2_O = diethylether; 1,2-PG = 1,2-propylene glycol; DMSO = dimethylsulfoxide; ETG = ethylene glycol; BnBz = benzyl benzoate; PEG = polyethylene glycol; HOAc = glacial acetic acid; n.r. = not reported; DER = drug-to-extract weight ratio during extraction; Δ = elevated temperature; GL = glycerol; OPEO = orange peel essential oil; CC = choline chloride; NADES = natural deep eutectic solvents; Lys = *L*-lysine; sc = supercritical; scE = supercritical extraction, usually a batch-type extraction process with scCO_2_ as the ES; HP-β-CD = hydroxypropyl-β-cyclodextrin; CAAE = a complexing agent-assisted extraction; MAE = microwave-assisted extraction; EM = emulsifier; PS = polysorbates, PECO = polyoxyethylene castor oil derivatives; LE = lecithin; MG = monoglycerides; PVA = polyvinyl alcohol; PEO = polyethylene oxide; PAA = polyacrylamide; cPAA = cross-linked PAA, CMC = sodium carboxymethyl cellulose; GG = guar gum; XG = xanthan gum; cPA = cross-linked polyacrylate; HPE = high pressure extraction; VRHE = vacuum resistive heating extraction.

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
