# Peer review of "Propolis Extract and Its Bioactive Compounds—From Traditional to Modern Extraction Technologies"

_molecules, 2021, doi:10.3390/molecules26102930_

Round 1

Reviewer 1 Report

This manuscript is a review dedicated to the bioactive molecules of propolis, their mechanism of action, and the extraction of propolis. The mechanisms of biological action of individual constituents and of propolis extracts are compared. The review could be useful, there are however several minor points which need additional attention by the Authors, as follows:

  1. The sentence in lines 39 – 41: Please rephrase, resins and pollen are not compounds.
  2. Line 41: Wax is also of animal origin.
  3. Line 55: Would be good to add a photo of a propolis trap.
  4. Lines 72 – 73: Brazilian red and brown propolis are quite different, please clarify.
  5. Line 84, geopropolis: There are many stingless bee species which do not produce geopropolis but just propolis. Geopropolis is a mixture of resins, wax and clay/soil.
  6. Line 118: It is necessary to mention at that point one of the most studied bioactive constituents of poplar type propolis - CAPE, an discuss its properties in more detail then the paragraph in lines 216 – 222.
  7. The extraction of propolis has recently been reviewed: Bankova et al. Journal of Apicultural Research, https://doi.org/10.1080/00218839.2021.1901426
  8. Lines 368 – 376: Funari et al. worked with green Brazilian propolis, but NADES have been used also for poplar type propolis.
  9. In Chapter 3. Technologies for propolis extraction, the Authors should mention extraction methods applying high pressure.
  10. In Table 1, it would be more informative and useful to the reader if the Authors include a specific column indicating the method used, e.g. maceration, MAE, etc.
  11. The Authors use many times the expression “molecular mechanisms of propolis” or “of propolis extracts”. This expression is somewhat confusing, these are molecular mechanisms of Action of propolis.

Author Response

We would like to thank you for finding time to read our paper and write us very constructive comments and suggestions. Naturally, we put the maximum effort into addressing all the suggestions to improve the paper's quality. Since we did a major revision, there are many changes in the original file. We uploaded the document with track changes and *all markup*, so that all these changes are visible. (The Line numbers reported here are the numbers from text with track changes – so they will not match when all changes are accepted.)

Here we note every change as suggested by reviewer:

This manuscript is a review dedicated to the bioactive molecules of propolis, their mechanism of action, and the extraction of propolis. The mechanisms of biological action of individual constituents and of propolis extracts are compared. The review could be useful, there are however several minor points which need additional attention by the Authors, as follows:

The sentence in lines 39 – 41: Please rephrase, resins and pollen are not compounds.

Line 41: Wax is also of animal origin.

Thank you, you are correct.

    Line 45 – We changed this part to:“ In general, propolis consists of plant resins and essential oils, beesswax, and pollen. Organic compounds that have been identified in propolis are: polyphenols, terpenes, esters, amino acids, vitamins, minerals, and sugars [1,2].“ We deleted the sentence: „However, there are also compounds of animal origin, like trace amounts of proteins (enzymes) from honey bee origin. “

    We continue with:

    „Since plant material comes in contact with the honey bee’s digestive system, esp. saliva, before being incorporated in the compartments of the beehive, propolis is considered to be an animal-derived product.“

Line 55: Would be good to add a photo of a propolis trap.

    Thank you, an excellent suggestion! We added a figure of collector traps for propolis from our archive (after line 63).

Lines 72 – 73: Brazilian red and brown propolis are quite different, please clarify.

    Of course, and there was an error regarding brown propolis. We corrected this to:

    „Brazilian red propolis is characterized by isoflavone formononetin (3), and isoliquiritigenin (4) [4]. while Cuban red propolis is rich in by polyisoprenylated benzophenones like nemorosone (5a,b) [16].“

    We also changed the reference No. 4, with the a more suitable one.

Line 84, geopropolis: There are many stingless bee species which do not produce geopropolis but just propolis. Geopropolis is a mixture of resins, wax and clay/soil.

Correct, we have deleted the sentence Line 81 (now a Line 99), and changed to: „Geopropolis is a slightly different type of bee glue, a mixture of resins, wax, clay, or soil. It is produced by a stingless bee from genus Melipona.“

Line 118: It is necessary to mention at that point one of the most studied bioactive constituents of poplar type propolis - CAPE, and discuss its properties in more detail then the paragraph in lines 216 – 222.

    We agree – this is an excellent suggestion. We wrote a new, expanded paragraph on CAPE and added new references accordingly: from line 238 to line 286.

The extraction of propolis has recently been reviewed: Bankova et al. Journal of Apicultural Research, https://doi.org/10.1080/00218839.2021.1901426

    Thank you, this helped us a lot - it is vital work. So now we cite this review several times in our paper. We added this as a reference No. 146.

Lines 368 – 376: Funari et al. worked with green Brazilian propolis, but NADES have been used also for poplar type propolis.

    Now this is line 438 to 447 – we have corrected this.

In Chapter 3. Technologies for propolis extraction, the Authors should mention extraction methods applying high pressure.

    We added the paragraph on HPE, from line 489 to 496, and the reference No. 145.

In Table 1, it would be more informative and useful to the reader if the Authors include a specific column indicating the method used, e.g. maceration, MAE, etc.

    We added a new column to the Table, with the extraction method used.

The Authors use many times the expression “molecular mechanisms of propolis” or “of propolis extracts”. This expression is somewhat confusing, these are molecular mechanisms of Action of propolis.

    We went through the text and changed this: line 29, 130, 139, 166, 220, 250, 292, 309

In summary, the main changes in the text are:

    The title has been changed.

    The abstract has been modified.

    We modified the first, third and fourth paragraphs of the Introduction

    The Figure 1 has been added.

    The descriptions below all molecular structures were deleted

    Six additional paragraphs on CAPE were added in part 2.

    A paragraph on HPE has been added in part 3.

    The column was added to the Table 1.

    New references were added: Ref. No. 4, 9, from 74 to 109, 145 and 146. Now there are 153 references.

Minor changes (typos, rephrasing, deletions, etc.) can be followed through the text.

A clear, easier-to-read version of the text will be uploaded upon the feedback & according to your instructions.

Reviewer 2 Report

In the manuscript titled “Bioactive molecules from propolis – from tradition to pharmaceutical innovation” by Jelena Šuran and colleagues, they have reported novel technologies to belong to an open research area that needs further effective solutions in terms of well-standardized liquid and solid extracts, which would be reliable in their pharmacological effects, environmentally friendly, and sustainable for production.  I have few concerns regarding the present manuscript.

-Items 1 and 2 are well-written and structured. The authors give a good “state of the art” according to the propolis key ingredients, extracts, and polyphenols.
-The authors have mentioned in the title “new technologies”, however in section 3 the authors have reported only general information, see Table 1
-Information about bioactive molecules from propolis need more information, and those molecules could be written again in the innovative technologies
-A new section, maybe section 4 are required with pharmaceutical innovation, or even products with propolis are required
-More information about polyphenols is required

Author Response

We would like to thank you for finding time to read our paper and write us very constructive comments and suggestions. Naturally, we put the maximum effort into addressing all the suggestions to improve the paper's quality. Since we did a major revision, there are many changes in the original file. We uploaded the document with track changes and *all markup*, so that all these changes are visible. (The Line numbers reported here are the numbers from text with track changes – so they will not match when all changes are accepted.)

Here we note every change as suggested by reviewer:

Review #1
-Items 1 and 2 are well-written and structured. The authors give a good “state of the art” according to the propolis key ingredients, extracts, and polyphenols.
-The authors have mentioned in the title “new technologies”, however in section 3 the authors have reported only general information, see Table 1
-Information about bioactive molecules from propolis need more information, and those molecules could be written again in the innovative technologies
-A new section, maybe section 4 are required with pharmaceutical innovation, or even products with propolis are required
-More information about polyphenols is required

Thank you for your helpful comments.

  • We agree that some additional information on bioactive molecules was necessary, so we increased information on caffeic acid phenethyl ester (CAPE), a representative & extensively studied polyphenol from poplar-type propolis (lines 239 and 286). We updated the references accordingly, Ref No. 74 to 109..
  • Also, we added an innovative method of propolis extraction in the last part, a high-pressure extraction as well as references No. 145 and 146. Thus we updated the Table with more details (extra column) on the methods of extraction.
  • The reviewer addressed an interesting point. The second part of our paper was about modern extraction technologies necessary to obtain propolis as an active pharmaceutical ingredient (API), and not about pharmaceutical innovations and products- this would be too broad, but an excellent idea for the next paper.
  • This is the main reason we changed the paper's title to:“ Propolis extract and its bioactive compounds– from traditional to modern extraction technologies“, to make it more precise regarding the content.

Reviewer 3 Report

I am pleased to read the manuscript presented for review. It should be said that this is a comprehensive knowledge review of propolis. The manuscript is well written. However, the authors are asked to complete the information in the abstract, which will allow the reader to get acquainted not only with the properties of the bee product, but above all with the information that will tell what was included in the article.
In the introduction, information should also be added that will reveal what the manuscript brings new in relation to other articles already published.

Author Response

We would like to thank you for finding time to read our paper and write us very constructive comments and suggestions. Naturally, we put the maximum effort into addressing all the suggestions to improve the paper's quality. Since we did a major revision, there are many changes in the original file. We uploaded the document with track changes and *all markup*, so that all these changes are visible. (The Line numbers reported here are the numbers from text with track changes – so they will not match when all changes are accepted.)

Here we note every change as suggested by reviewer:

I am pleased to read the manuscript presented for review. It should be said that this is a comprehensive knowledge review of propolis. The manuscript is well written. However, the authors are asked to complete the information in the abstract, which will allow the reader to get acquainted not only with the properties of the bee product, but above all with the information that will tell what was included in the article.
In the introduction, information should also be added that will reveal what the manuscript brings new in relation to other articles already published.

We appreciate your suggestions. We modified the abstract (line 21 to line 37) and the end of the  Introduction (line 75 to line 80). Thus, we added two references, No 4 and 9. We hope these versions will be better.

Reviewer 4 Report

This review describes the bioactive components from propolis. In addition, the extraction method of propolis also demonstrated. However, the authors pay more attention on the design of the whole manuscript, especially in the second part of the manuscript “Propolis types, key molecules, and their biological activities”. It is a bit mess due to some repeat texts or unnecessary texts. I suggest the author re-organize that part to make it clearer and more concise. In addition, the references cited in the manuscript can not match the number of references in the refence list. Except the above mentioned, there are still other text and format errors. Some of the English is rather awkward in places. Hereby, I suggest the author should read the manuscript again carefully.

I also encourage the authors to address the comments below.

P1L35 The English is awkward and please revise.

P1L37 change the word “problem”

P2L63 remove “,”

P2L65-P7

The author tried to overview the composition of bioactive compound and biological activities of extracts, but some texts are not necessary or repeat again. Hereby, I suggested the author re-organize the composition of marker and biological activities, respectively.

P3L87 gallic acid and ellagic acid

P3L91 analyzed to found.

P3L95 revised the last sentence.

P8L279 revised the first sentence.

P9L281 delete so-called

P9L282 what does the “parts” mean? “times”

P9L287 revised “After the maceration, the liquid extract is isolated by separating undissolved propolis residues by simple filtration through filter paper or a suitable mesh screen”

P8L311 you already have API in front, so delete active pharmaceutical ingredient

P8L322 revised the sentence.

P9L322 re-write

P9L373 re-write

Author Response

We would like to thank you for finding time to read our paper and write us very constructive comments and suggestions. Naturally, we put the maximum effort into addressing all the suggestions to improve the paper's quality. Since we did a major revision, there are many changes in the original file. We uploaded the document with track changes and *all markup*, so that all these changes are visible. (The Line numbers reported here are the numbers from text with track changes – so they will not match when all changes are accepted.)

Here is our reply point by point:

This review describes the bioactive components from propolis. In addition, the extraction method of propolis also demonstrated. However, the authors pay more attention on the design of the whole manuscript, especially in the second part of the manuscript “Propolis types, key molecules, and their biological activities”. It is a bit mess due to some repeat texts or unnecessary texts. I suggest the author re-organize that part to make it clearer and more concise. In addition, the references cited in the manuscript can not match the number of references in the refence list. Except the above mentioned, there are still other text and format errors. Some of the English is rather awkward in places. Hereby, I suggest the author should read the manuscript again carefully.

This feedback has been very valuable to us. We went through the manuscript carefully, and we made many corrections in the text for the sake of brevity & clarity. The tracked changes can be seen with *All markup* function.

I also encourage the authors to address the comments below.
P1L35 The English is awkward and please revise.

Corrected to: „Propolis is one of the most known honey bee products, used in folk medicine since ancient times for its numerous health effects.“ And added a new sentence: „Nowadays, it is starting raw material for manufacturing various extracts that can be used as active pharmaceutical ingredients (APIs).“

P1L37 change the word “problem”

Changed „problem“ to „crisis“

P2L63 remove “,”  - Removed

P2L65-P7

The author tried to overview the composition of bioactive compound and biological activities of extracts, but some texts are not necessary or repeat again. Hereby, I suggested the author re-organize the composition of marker and biological activities, respectively.

We agree with the reviewer. First, we deleted the parts that were repeating. For example,

  • Line 53: „as propolis is rich in bioactive molecules with antimicrobial, antioxidant, and immunomodulatory activity“
  • in line 153: „Polyphenols also induce the transcription of antioxidant enzymes“
  • line171: „ such as antioxidant, anti-inflammatory, anticancer and antiviral activities“,
  • line 206, the part: „antioxidant, anti-inflammatory, neuroprotective, antimicrobial, and anticancer activity“,
  • Line 208: „protects the cell against oxidative stress...“
  • Line 307: „Antioxidant activity is one of the most relevant effects of propolis“:
  • Line 336: „The glycolic extract (1,2-PG) had the highest total polyphenol content (TPC) (81.2 ± 3.7%), followed by hydroalcoholic extract (69.7 ± 2.0%). The lowest TPC was obtained in powder, a micronized sample composed of propolis with a minimum of 12% total polyphenols supported by sucrose and silicon dioxide (16.5 ± 0.8%), and oily extract (24.0 ± 1.4%). Nevertheless, all extracts had comparable antioxidant activity.“ – since this is commented in the last part of the manuscript.

After deleting the repeating parts, the text was clearer, and we also rephrased sentences or re-organized some paragraphs (2.1.3. Galangin). We sincerely hope this will be easier to read.

P3L87 gallic acid and ellagic acid

  • Done

P3L91 analyzed to found.

  • Done, changed.

P3L95 revised the last sentence.

  • Changed to „This kind of polyphenol content is typical for poplar propolis.”

P8L279 revised the first sentence.

  • Revised to „The crude propolis is traditionally extracted with extraction solvents (ES) consisting of various ethanol (EtOH) and water mixtures. Typically, 25-60% v/v aqueous EtOH is used as the ES at room temperature (r.t.), yielding the propolis tincture.“

P9L281 delete so-called

  •  

P9L282 what does the “parts” mean? “times”

  • It refers to quantity, in volume or weight. For the sake of clarity, we now express it as ratio (line 351): „It is conducted by the addition of aqueous EtOH onto the crude propolis chunks at the weight (propolis) to volume (ES) ratio of 1 : 3-20, most commonly 1 : 5-10.“

P9L287 revised

  • Revised to: “ After the maceration, the liquid extract is separated from undissolved propolis residues by simple filtration“, line 355

P8L311 you already have API in front, so delete active pharmaceutical ingredient

  • Deleted

P8L322 revised the sentence.

P9L322 re-write

  • Line 389, Re-written to:

“Water is more polar solvent than 60-80% aqueous EtOH, and it extracts more polar propolis compounds. The efficacy of water as the propolis ES could be increased by the use of:

  • repeated extraction at cold processing conditions [131],
  • elevated extraction temperatures ranging from 50-95°C [129,130,132, 133],
  • Soxhlet extraction technique [128], or,
  • ultrasonic-assisted extraction (UAE) [133].”

P9L373 re-write

  • Line 445, Re-written to:
  • “Alternatively, aqueous solution of amino acid L-lysine (Lys; 10%) is relatively effective ES for propolis, yielding the corresponding liquid extract with about 50% of the components concentration obtained with aqueous ethanol (70%)”

Round 2

Reviewer 2 Report

Thank you to the authors for taking into account my previous comments

Author Response

We kindly thank reviewer for effort.  English language and style are carefuly checked.

Reviewer 4 Report

In abstract, L20, "These bioactive molecules, polyphenols," Is not polyphenols being one member of bioactive molecules? 

L24, propolis? should be extracts?

L106 either "gallic acid and ellagic acid" or "P3L87 gallic and ellagic acids"

The paragraphs in the manuscript are not in the correct format (like L606 back to normal)

The authors should double check the whole text to ensure only one abbreviation for each word. There are some texts in multiple copies, such as EtOH, ethanol. 

Author Response

We kindly thank reviewer for valuable comments, we did our best to solve the questions raised by reviewer and hope that manuscript will now meet criteria to be published.